# Effect of Circadian Rhythm Disturbance on the Human Musculoskeletal System and the Importance of Nutritional Strategies

**DOI:** 10.3390/nu15030734

**Published:** 2023-02-01

**Authors:** Norsham Juliana, Liyana Azmi, Nadia Mohd Effendy, Nur Islami Mohd Fahmi Teng, Izuddin Fahmy Abu, Nur Nabilah Abu Bakar, Sahar Azmani, Noor Anisah Abu Yazit, Suhaini Kadiman, Srijit Das

**Affiliations:** 1Faculty Medicine and Health Sciences, Universiti Sains Islam Malaysia, Nilai 71800, Malaysia; 2Faculty of Health Sciences, Universiti Teknologi MARA, Bandar Puncak Alam 42300, Malaysia; 3Institute of Medical Science Technology, Universiti Kuala Lumpur, Kajang 43000, Malaysia; 4Anaesthesia and Intensive Care Unit, National Heart Institute, Kuala Lumpur 50400, Malaysia; 5Department of Human & Clinical Anatomy, College of Medicine & Health Sciences, Sultan Qaboos University, Al-Khoud, Muscat 123, Oman

**Keywords:** chrononutrition, biological clock, circadian, skeletal muscle, metabolism

## Abstract

The circadian system in the human body responds to daily environmental changes to optimise behaviour according to the biological clock and also influences various physiological processes. The suprachiasmatic nuclei are located in the anterior hypothalamus of the brain, and they synchronise to the 24 h light/dark cycle. Human physiological functions are highly dependent on the regulation of the internal circadian clock. Skeletal muscles comprise the largest collection of peripheral clocks in the human body. Both central and peripheral clocks regulate the interaction between the musculoskeletal system and energy metabolism. The skeletal muscle circadian clock plays a vital role in lipid and glucose metabolism. The pathogenesis of osteoporosis is related to an alteration in the circadian rhythm. In the present review, we discuss the disturbance of the circadian rhythm and its resultant effect on the musculoskeletal system. We also discuss the nutritional strategies that are potentially effective in maintaining the system’s homeostasis. Active collaborations between nutritionists and physiologists in the field of chronobiological and chrononutrition will further clarify these interactions. This review may be necessary for successful interventions in reducing morbidity and mortality resulting from musculoskeletal disturbances.

## 1. Introduction

Human physiological functions are highly dependent on the regulation of the internal circadian clock. The complete biological clock includes the master clock in the hypothalamus’s suprachiasmatic nucleus (SCN) and the peripheral clocks. Both clocks must work in symphony, with the SCN playing the role of the timekeeper of the whole body [1]. The body’s musculoskeletal system is part of the peripheral clocks; hence, the system relies on the SCN regulation of neural and endocrine pathways to work synchronously with other systems [2].

Despite the reliance on the light that entrains the sleep and wake cycle, the peripheral clock in the musculoskeletal system also depends on the regulation of the 24 h schedule of feeding and physical activities. These scheduled regulations that influence the musculoskeletal system’s circadian clock are independent of the SCN clock. Zhang et al. found that there are approximately 1600 circadian genes expressed in the skeletal muscle [3]. Clock-controlled genes (CCGs) that are important for the circadian maintenance of skeletal muscle and specific to the musculoskeletal system include Myod1, Atrogin (Fbxo32), Mef2, and Glut4 [3,4].

Skeletal muscle constitutes approximately 40% of total body weight and contains 50 to 75% of the body’s total proteins. Therefore, it consists of the largest collection of peripheral clocks in the human body. Metabolically, the skeletal muscle is responsible for maintaining homeostasis in the balance between energy intake and utilisation [5]. Equilibrium is always achieved because of the unique musculoskeletal capabilities to adapt based on the body’s demand (physical activity) and supply (nutrient). This flexibility allows the skeletal muscle system to decide on the usage and storage of energy required for the body based on the 24 h schedule of fasting–feeding and resting–active state [6,7].

Similar to skeletal muscle, the circadian rhythm that regulates bone functions also depends on the cues from scheduled feeding and fasting. Serum *C*-telopeptide fragments of collagen type 1 (S-CTX), a serum marker on bone resorption, showed that its daily rhythm is highly influenced by the fasting–feeding schedule. Its serum level is higher in the early morning and reaches a peak at around 05:00–08:00, whereas in the late afternoon, from 12:00 to 16:00, the serum level reaches its lowest level. An apparent change in diminished diurnal rhythm of the marker is seen in fasting [8]. Besides that, the diurnal rhythm also varies with changes in the intake of glucose, protein, and fat [9]. Due to the influence of the circadian rhythm on the musculoskeletal system, those working in shifts and with circadian disruption are at increased risk of bone fractures and osteoporosis [10,11].

Both central and peripheral clocks regulate the interactions between the musculoskeletal system and energy metabolism. Thus, external factors such as feeding time and various nutrients mediate its regulation. This review discusses the disturbance of the circadian rhythm and its resultant effect on the musculoskeletal system. We also discuss the nutritional strategies that are potentially effective in maintaining the system’s homeostasis.

## 2. Methods

The search strategy was developed collaboratively with the research team. A time frame for publication access was set to focus on the recent studies on circadian rhythm and the resultant effect on the musculoskeletal system of the body, together with the related nutritional strategies (from 2012 to 2022). The systematic search was performed in electronic databases using standardised search terms specific to the needs of each respective database and revised by the research team. The databases selected were Scopus, Pubmed, EBSCOHost, and Google Scholar. The search was conducted with MeSH (Medical Subject Headings in PubMed) terms including “circadian rhythm”, “biological clock” “musculoskeletal disease”, “musculoskeletal abnormalities”, “diet”, “nutrition”, and “nutritional status”. Boolean operators (AND, OR, AND NOT) were used to combine search terms within related keywords. An additional search was performed using modified search terms if search terms were incomplete. For this review, articles were restricted to English publications.

## 3. Human Circadian Rhythm

Humans display a circadian rhythm of an approximately 24 h cycle. The word “circadian” is derived from the Latin phrase circa diem, which means about a day. The rhythm may vary from one human to another depending on their lifestyle and environment. In a healthy human, the circadian rhythm influences major physiological functions, including the 24 h biological cycle of behavioural, physical, and mental changes. More specifically, the major physiology functions of circadian rhythm involve, among others, the sleep–wake cycle [12,13], blood pressure [14,15], body temperature [16,17], hormonal homeostasis [18], prothrombotic factor regulation [19], and hormone release in the digestive system to match eating habits [20]. Besides these, circadian rhythm is also important in the promotion of good metabolic balance, haemostasis, alertness or functional cognition, optimum cell repair, a healthy body, and healthy brain development, in addition to the boosting of the immune system [21]. Figure 1 illustrates the 24 h circadian rhythm and human body response.

A group of nerve cells forming the suprachiasmatic nucleus (SCN) in the anterior hypothalamus play roles in regulating the circadian rhythm. The SCN is set by the circadian rhythm clock, which receives signals through the optic nerve (input) connected to the eyes. The molecular mechanism of circadian rhythms involves the circadian oscillation mechanism within cells that is controlled by a genetically encoded molecular clock. The endogenous biological pacemaker primarily controls the circadian rhythm. Many people believed that the circadian rhythm was solely controlled by the SCN, acting as the master clock. However, recent studies discovered that the circadian rhythm is also controlled by peripheral cells, such as those cells in the cardiovascular system, which possess a similar circadian clock [22,23]. As a result, both the central clock (SCN) and the peripheral clock (non-SCN) work together to provide the essential rhythmic function of the body’s systemic cues. 

A gene mutation encoding a protein or unnatural light exposure may disrupt the circadian rhythm [24]. A chronically disrupted circadian rhythm can cause conditions such as jetlag, fatigue, insomnia, and other chronic medical conditions such as cancer, diabetes, and cardiovascular disease [14,25]. Period (PER) was the first gene identified in the circadian rhythm. Other core genes were later discovered, such as the transcription factors CLOCK-BMAL1 in the main loop which induces negative regulators, which are the important protein encoding genes, Period (PER) and Cryptochrome (CRY) proteins [26]. Other core genes, such as REVERBs/ROSs and Albumin D-site-Binding Protein (DBP) have also been found to be involved in the circadian rhythm [27]. At the molecular level, the circadian clock comprises various sets of transcription factors that lead to the autoregulation of transcription–translation feedback loops (TTFLs). The regulation represents the core mechanism of a mammal’s circadian clock. Figure 2 represents the simplified version of molecular interaction of cell cycle components with the circadian core clock. 

### 3.1. The Biological Clock

The mammalian circadian rhythm is controlled by the biological “clock gene” to produce the clock protein [29]. The central component of the molecular mechanism for naturally occurring circadian rhythm involves the interlocking of transcription–translation feedback loops (TTFL) with other core genes [27,30,31,32]. This component controls the oscillation of cells and tissues on their built-in (endogenous) and adjustable (entrainable) adaptation to external factors, such as daily regular environmental changes, i.e., the sleep–wake cycle [13,22]. Additionally, the central component controls melatonin secretion to control sleep patterns [33,34]. The molecular mechanism and genes involved in the circadian rhythm’s biological clock in mammals are primarily controlled by light exposure. The alerting system follows a pathway called the photic neural input pathway. In the pathway, light plays a role as a signal molecule that is detected by the retina in the presence or absence of light. The light alters the pineal gland secretion, and the eyes are closed involuntarily to avoid any excessive exposure to light [35,36,37]. These mechanisms are regulated by the circadian rhythm, thus explaining their importance in daily life. 

Moreover, there are pathways that serve as modulators for peripheral tissues. First, the pathways are activated by the signals detected by the photosensitive retinal ganglion cells, followed by the retinohypothalamic tract (RHT) pathway, which is then connected to the SCN, which then releases the output instruction via the paraventricular nucleus (PVN) and superior cervical ganglia (SCG). Then, the output is delivered to the targeted endocrine gland, such as the pineal gland, for melatonin inhibition and secretion [35]. Reduced blood pressure, lower levels of melatonin secretion, and an increase in cortisol are the factors that cause the body to naturally feel awake in the early morning, as the body has optimal cardiovascular efficiency, compromised muscle strength, and increased brain performance. 

### 3.2. How Is Circadian Rhythm Measured?

Measuring circadian rhythm is important to assess an individual’s sleep–wake cycles, especially in patients with circadian rhythm sleep–wake disorder (CRSWD). A variety of measures mainly involving physiological function and instruments or objective tools can be used to assess circadian rhythm. Melatonin level, core body temperature, and the rest–activity cycle are examples of physiological functions. Melatonin level is commonly used due to its accessibility, as it can be detected through plasma and/or saliva. This allows for frequent sampling. However, the assay is not a standard used in laboratories due to its cost and high turnaround time (7–10 days) [38]. Dim light melatonin onset (DLMO) is a marker commonly used and recommended to confirm the diagnosis of sleep–wake disorder [39]. Aside from that, core body temperature has previously been used to assess circadian timing due to the faster results obtained compared to a melatonin assay [40]. Previously, this method was less convenient as a probe had to be inserted into the rectum, but technology advancements have led to devices that can be swallowed to transmit real-time data through the digestive tract, as well as measure skin temperature [41]. The rest–activity cycle can be used to measure circadian timing. Wrist actigraphy measures the sleep–wake cycle and light level to track the timing of rest–activity. Multiple features are available in commercially available wrist-worn activity monitors, which are believed to be beneficial in estimating sleep–wake cycles that are associated with sleep–wake disorders [39]. Assessing light levels was also found to have a relationship with mood, sleep quality, and health-related outcomes [42,43]. 

Besides physiological function, objective measurements using tools such as sleep diaries and chronotype questionnaires were also commonly used due to their feasibility in population studies. Clinicians can use sleep logs to characterise patterns of sleep–wake. The diaries also included information about medications, alcohol or caffeine consumption, naps, and exercised activity, which was useful in assessing the pattern to improve circadian disturbance in patients with sleep disorders [39]. Circadian chronotype is an individual’s preferred time to perform daily activities. As an alternative to timing the circadian rhythm, questionnaires are frequently used tools. The questionnaires fit well in healthy control populations, but evidence was lacking in patient populations [44]. Examples of the questionnaires include the Morning–Eveningness Questionnaire (MEQ), Munich Chronotype Questionnaire (MCTQ), and Munich Chronotype Questionnaire for Shiftworkers (MCTQShift) [45].

### 3.3. Effect of Circadian Rhythm on the Body

The circadian rhythm holds a crucial regulatory role in human physiology by regulating many different body systems, including hormone production, glucose homeostasis, the central nervous system, cardiovascular maintenance, and lipid metabolism, to name a few. All these systems are interrelated and dependent on each other, thus imposing the role of circadian rhythm on human physiology. Since the SCN regulates body temperature, neuronal activities, hormone secretion, and musculoskeletal systems, it is deduced that disruption in the normal circadian rhythm can impair the regulated systems. The SCN, in turn, is majorly dependent upon light. At a molecular level, the circadian rhythm is regulated by the transcriptional and translational regulation feedback loop of core clock genes, which include Bmal1 (Bain and muscle ARNT-like 1), CLOCK, Per1 (Period1), Per2, Cry1 (cryptochrome1), and Cry2. Thus, the CLOCK: Bmal1 regulators are an integral part of SCN regulation and are directly affected by external factors such as sleep quality, diet, and exercise. Sleep disturbances, for example, have been linked to neuro-disorders such as attention-deficit hyperactivity disorder (ADHD) [46]. ADHD studies showed that children with delayed melatonin secretion are predisposed to ADHD [47]. In adults, impaired CLOCK gene expressions caused by disturbed sleep have been reported to perturb the diurnal rhythms of endocrine secretion [48]. Thus, compromised sleep has been correlated with ADHD in adults. 

Apart from melatonin secretion, another example of hormonal regulation by the SCN is the maintenance of the hormone leptin. Leptin increases the activation of the sympathetic nervous system and increases thermogenesis by stimulating the thyroid hormones [49]. Studies show that the peak of leptin secretion occurs during the night. In a study by Buonfiglio et al., lack of sleep resulted in long-term resistance to leptin, and led to reduced metabolism and increased adipose tissue in rats [50]. Conversely, numerous studies demonstrated that rats supplemented with oral melatonin showed increased body weight loss [51]. Such studies have also shown increased weight loss in post-menopausal women [52]. In this study, 81 women between the ages 25 and 65 years were subjected to melatonin administration and a balanced diet to improve their sleep cycle. The tested subjects reported weight loss and improved sleep quality after 24 weeks. On top of obesity, circadian rhythms have also been implicated in other metabolic illnesses, including type 2 diabetes and several neurological disorders.

Numerous studies have also shown a positive correlation between circadian rhythm and glucose homeostasis. Significant changes in transcription factors in liver genes such as forkhead box O1 Foxo1 and peroxisome proliferator-activated receptor alpha, Pparα, increased. Such oscillations restored homeostasis and increased insulin sensitivity to accommodate glucose metabolism in disrupted sleep cycles [53]. However, glucose homeostasis is significantly affected in diabetic conditions and by chronic circadian disruption. In clock-disrupted clock-gene Bmal1-knockout mice, insulin secretion was reduced, and they were prone to obesity. In shift work mice, the liver clock genes and glucose metabolism-related genes oscillated. Restored Bmal1 genes rescued the phenotype, thus highlighting the role of circadian genes in insulin expression [54].

Interestingly, recent work by Xu et al. (2022) shows that even in the prediagnostic phase of diabetes mellitus, the mild disruption of circadian rhythms and daily rest–activity impairs glucose homeostasis and thus predisposes one to diabetes mellitus [55]. Since metabolism is crucial in cellular turnover, perturbations in the circadian cycle can induce metabolic disorders such as non-alcoholic fatty liver disease [56]. Non-alcoholic fatty liver disease is also commonly associated with the loss of skeletal muscle mass, or sarcopenia. Such a correlation indicates the relationship between musculoskeletal systems and metabolism [57]. Both the skeletal and muscle systems are implicated in the body’s locomotor activity and nutrition homeostasis which, in turn, maintain tissue mass and metabolism.

### 3.4. Effect of Circadian Rhythm on the Musculoskeletal System

The CLOCK: Bmal1 circuit also regulates the circadian regulation of the musculoskeletal system (Figure 3). In osseous structures, a synchronous circadian rhythm promotes healthy cartilage and bone formation [58,59,60]. The regulation of bone formation by osteoblasts and absorption by osteoclasts is maintained by glucose homeostasis and parathyroid hormones [61], all of which are regulated by circadian rhythms [62,63]. A study showed that a combined high-fat diet and circadian rhythm disruption lead to a diabetic skeletal phenotype in mice [24]. The disruption of the Bmal1 expression system has been shown to compromise bone and cartilage formation. Transcriptomics in Bmal1-knockout mice showed degenerating knee cartilage due to chronically impaired circadian clock activity [64]. Bmal1 disruption also perturbs glucose homeostasis in the musculoskeletal system. The disruption of Bmal1 expression resulted in a muscle-specific loss by disrupting muscle triglyceride biosynthesis and accumulating bioactive lipids and amino acids [65]. In 9-month-old Bmal1-knockout mice, the absence of Bmal1 expression led to muscle atrophy, reduced body weight in healthy mice, weak skeletal muscles, and structural muscle pathologies [66]. Muscle pathologies were also shown to occur due to impaired glucose metabolism. Numerous studies show that diurnal oscillations cause insulin resistance [67] and diminish the expression of the glucose transporter 4 (GLUT4) [68,69]. Another study showed that GLUT4 expression is increased with exercise and increases insulin sensitivity and glucose absorption, thereby promoting muscle health [70]. The musculoskeletal system is heavily involved with locomotion, feeding, and sleep quality. Thus, intervention through the chrono diet, sleep, and exercise can be used to maintain and sustain healthy bone and muscle growth [71].

### 3.5. Role of Skeletal Muscle Circadian Clock in Lipid and Glucose Metabolism

It is generally well-established that clock genes regulate the circadian rhythm which is associated with glucose and lipid metabolism [72]. Hence, disturbances in the circadian rhythm, particularly due to shift work, impair lipid and glucose homeostasis, ultimately impacting human health [73]. The circadian-regulating process has been recognized as one of the factors that is linked to the oscillation of glucose metabolism via the hepatocyte circadian clock. Due to this, shift work in the long term has been shown to reduce tolerance towards glucose [73]. Various genes which regulate glucose metabolism, such as Gys2 [74], PEPCK [75], CRY1, and CRY2 [76], have been studied to be associated with cellular circadian rhythm. 

An experimental tissue-specific Bmal1 knockout (KO) model has unravelled the critical role that peripheral oscillator circadian clocks have on glucose metabolism, especially in the pancreas, liver, and skeletal muscle [77]. Studies using the skeletal muscle-specific Bmal1 KO have shown that the muscle clock is essential in anticipating the transition from resting to active phase at awakening and promoting glucose uptake and metabolism, as the predominant fuel for skeletal muscle [78].

Dyar et al. revealed that the muscle clock via the Bmal1 and REV-ERBα is a key factor in regulating various normal physiological process such as muscle energy homeostasis [65]. This occurs through the direct activation of transcriptional activities that promote lipid storage, insulin sensitivity, and glucose metabolism, while simultaneously inhibiting lipid oxidation and protein catabolism. Disturbances in the muscle clock genes results in increased lipid oxidation and protein turnover, which culminates in a metabolic inefficiency condition. 

A short-term controlled experimental circadian misalignment study using a shift work clock revealed a disruption in skeletal muscle lipidome, which led to insulin resistance [79]. In an earlier experiment on circadian misalignment, skeletal muscle insulin sensitivity was decreased by 23% in contrast to the normal circadian condition [80]. This suggests that circadian misalignment disrupts skeletal muscle lipid metabolism, which contributes to the occurrence of muscle insulin resistance, eventually becoming a precursor to type 2 diabetes mellitus [79]. Inevitably, alterations in the circadian rhythms and the genes that control them affect normal physiological metabolism, eventually resulting in various metabolic disorders [81].

## 4. Circadian Rhythm and Bone Health

Bone health is dependent on the dynamic balance between bone formation by osteoblasts and bone resorption by osteoclasts. Biological factors such as hormones, chronobiological properties, and cytokines, as well as mechanical factors such as load and stressor, influence these activities. Circadian rhythm has been postulated to have an effect on bone remodelling. Our body needs to keep up with its circadian clock to stay healthy. Those who have an altered or reversed circadian rhythm, such as shift workers, were found to have pre-osteoporotic features such as reduced bone mass and density, altered bone microstructure, and decreased bone strength [82]. Animal studies are also in accordance with this finding, in which circadian gene knockout mice showed abnormal bone remodelling [83]. 

Previous studies have reported that clock genes are involved in osteoblasts and osteoclasts’ activities as well as cartilage formation [84]. The serum level of calcium and bone markers such as alkaline phosphatase, *C*-telopeptide of type I collagen (CTX), tartrate-resistant acid phosphatase (TRAP), and osteocalcin showed diurnal variation, which peaked in late night or early morning about 15–20% higher than the nadir value, which occurs in the afternoon [85]. Most bone cells, such as osteoblasts, osteoclasts, and osteocytes, have been shown to express circadian clock genes. The main clock genes that regulate circadian rhythm and bone health are Bmal1, Bmal2, Clock, Cry, and Per [8]. Bmal1 plays a vital role in regulating bone remodelling by upregulating the expression of bone microRNAs, such as the nuclear factor of activated T cells and inhibiting the receptor activator of nuclear factor kappa–beta ligand (RANKL). RANKL binds to RANK on osteoblasts which later activate osteoclast differentiation. This stimulation of osteoclastic activity increases bone resorption. The overexpression of RANKL is associated with bone degenerative diseases such as osteoarthritis and rheumatoid arthritis. Bmal1-/- mice were found to have osteopenia. This is in accordance with the fact that Bmal1 is associated with the inhibition of RANKL. Hence, the absence of Bmal1 promotes bone resorption activity and leads to a decrease in bone mass and density [86]. Cry and Per, which are also the main clock genes, have a negative regulation factor for circadian rhythm. The mutation of these genes results in increased osteoblast proliferation and reduced bone loss. Cry and Per have low expression during the day and high expression during the night. This is in congruence with the expression of bone-related genes, where osteoprotegrin (OPG) expression rises during the day and falls during the night, and vice versa for RANKL [87]. This evidence is in accordance with a study by Samsa et al., which reported that Per- and Cry-deficient mice had low bone volume [88]. Correspondingly, continuous light exposure can lead to bone loss and an increase in inflammatory cytokines, which result in an increased risk of osteoporosis.

### Less Number of Hours of Sleep and Associated Bone Defects and Osteoporosis

The pathogenesis of osteoporosis includes an imbalance between bone resorption and bone formation activities as we age, an increase in inflammatory cytokines, a lack of sexual hormones, medications, and a shift in circadian rhythm. In recent years, it has been reported that disturbances in the dark–light cycle, which generate a condition similar to shift work or chronic jet lag, showed a trend in alterations in bone structure and bone loss [89]. Adequate hours of sleep are important for biological processes and to maintain optimum health. Biochemical bone markers such as CTX, osteocalcin, and ALP peak overnight, with a peak in the early morning, which suggests that bone remodelling is affected in those who experience sleep disturbances [90]. There have been many reports of increased fracture risks among shift workers. Sleep disturbances and circadian reduction can increase somnolence, which indirectly reduces self-vigilance towards environmental hazards and increases risk of falls and fractures [91]. 

In an osteoarthritis rat model, it was found that clock genes in the early stages of bone and cartilage degeneration were altered. Another animal study using rats with a 10-day sleep restriction showed a significant drop in osteoblast and increase in osteoclast activity [92]. Another study using a different method reported that mice exposed to continuous light for 24 weeks had decreased trabecular bone volume compared to mice exposed to normal light–dark cycles [59]. Their trabecular bone was reported to be porous, with an increase in inflammatory markers. These findings supported the correlation between sleep disturbance, circadian rhythm, and bone health. Studies have demonstrated that those who worked night shifts had lower bone density compared to those who worked day shifts [90,93,94]. It was speculated that the low bone density in night shift workers is due to an increase in cortisol and altered vitamin D status from sleep disturbances. 

A study conducted by Shan et al. reported a correlation between shift work and an increased risk of obesity and type 2 diabetes mellitus (T2DM) [95]. Energy expenditure is affected due to sleep deprivation, and circadian rhythm disturbances cause an increase in energy intake. As a result, they contribute to overweight and T2DM. There is a close link between bone health and glucose metabolism. High blood sugar levels are reported to inhibit calcium absorption and bone calcification, which lead to an increased risk of osteoporosis [96]. In T2DM mice, serum osteocalcin was reduced, and serum TRAP was significantly increased. Bone density and bone strength were also significantly reduced in those T2DM mice. Hence, we can corroborate that sleep disturbances affect glucose metabolism which, in turn, leads to bone loss. 

Diurnal variations in bone cells and bone markers as well as the knockout clock gene in animal models have clearly shown that circadian rhythmicity is vital for bone health. Nowadays, an increasing number of individuals experience sleep disturbances due to working shifts and jet lag due to constant travel. This contributes to an increased risk of osteoporosis, which could be attenuated by a novel intervention targeting the circadian timing system. The application of chronotherapy, which is a method of medication administration coordinated with the biological clock, is an effective intervention to improve the efficacy of osteoporosis treatment. The circadian clock gene can be a therapeutic target for researchers to study on the prevention and treatment of bone diseases. This indirectly reduces osteoporosis and fracture cases in the future. Table 1 describes a list of studies on bone disorders and their relationships with circadian rhythm.

## 5. Nutritional Strategies for Musculoskeletal Health

Nutritional status plays an important aspect in bone health. Malnutrition appears as a strong predictor of bone osteoporosis and impaired bone healing, especially in the elderly. Hence, a balanced diet rich in nutrients, vitamins, and minerals is essential for the prevention and treatment of bone disorders [112]. 

Besides bone health, muscle disorders that are usually associated with pain are also the leading causes of disability and decreased quality of life among adults worldwide. Their prevalence was recorded at approximately 30% (ranging from 13.5% to 47%) in 23 population studies conducted in 15 different countries (the United States, United Kingdom, Germany, Sweden, Denmark, Norway, Italy, Spain, The Netherlands, Kuwait, Japan, Austria, Malaysia, Brazil, and Cuba). As practitioners struggle to curb the problems, it is proposed that the modifiable factor related to nutrition be included in the assessment, advice, and intervention, which may provide a new perspective on the solution [113]. Multiple disorders of the musculoskeletal system are related to inflammatory factors; hence, they exert the sensation of pain. A diet containing protein, specifically soy protein, has been found to have a pain-relieving effect among an aging population with osteoarthritis [114]. Several studies have suggested that anti-inflammatory foods, such as those found in the Mediterranean diet (vegetables, fruits, fish, fish oils, legumes, olive oil, and plant-based proteins), are best for relieving the inflammation and pain associated with chronic musculoskeletal disorders and diseases [113]. On the contrary, an excessive fat diet was found to increase nociceptive responsiveness and enhance pain sensation in an animal study on knee arthritis [115]. 

Calcium and vitamin D are two established nutrients related to bone health. Deficiencies in calcium and vitamin D are major risk factors for osteoporosis and compromised bone repair [116]. An adequate intake of these nutrients is essential; however, the global prevalence of deficiencies is notably high [117]. Because calcium intake influences peak bone mass from a young age, it is critical to increase children’s consumption of dairy foods and calcium sources in food or supplements. Previous meta-analyses reported that calcium or dairy intervention in children has a significant effect on bone density [118,119]. Although it is uncertain whether these effects could reduce fracture, adequate intake of calcium at a young age is shown to be related to bone mineral accretion in adulthood as well. Recent research updates highlighted on the importance of chronotherapy as a strategy to improve the therapeutic efficacy of osteoporosis because bone metabolism demonstrates potent circadian regulations. In addition, the main regulators of calcium homeostasis, namely osteocalcin and vitamin D, also demonstrate a diurnal rhythm in humans. Thus, the appropriate timing for the supplementation of calcium may have a significant impact on bone resorption. There are studies suggesting that evening calcium intake may suppress bone resorption. On the contrary, no significant impact was found when the supplementation was given in the morning [120,121]. However, up to date, the optimal time for antiresorptive treatment is still scarce. Despite the fact that calcium supplementation and therapy based on the biological clock seems to be a promising strategy, more data from randomised clinical trials are needed prior to its implementation. 

Vitamin D on the other hand, is essential to maintain the concentration of calcium [122]. It can be obtained from sunlight exposure and food fortification. The deficiencies are worldwide issues which are mainly associated with bone diseases [123]. Eventually, a recent study among shift workers showed that low serum calcium levels might be associated with circadian dysregulation as well as sleep disturbance [124]. Individuals with irregular sleep schedules may obtain less sun exposure, which contributes to vitamin D deficiency. Due to inevitably low exposure to sunlight in certain populations owing to various limitations, there are suggestions for vitamin D supplementation. A few studies have suggested that the supplement be taken after meal to maximize absorption [125]. Similar to calcium supplementation, the intake of vitamin D supplements at a certain time of the day is the future direction of study to be explored. 

Nutritional strategies for bone health in circadian dysregulation population may include promoting an adequate intake of calcium and vitamin D from a young age for the promotion of bone development. It should be in accordance with physical activity for improvement in bone mineral density [126]. School-based nutrition intervention may be effective in implementing the strategy [127]. In adults, as they may be exposed to circadian regulation, a balanced diet emphasising calcium is necessary to maintain bone health. Fortified vitamin D foods, milk intake, and calcium supplementation are also suggested [128]. Although adult intervention is questionable in terms of feasibility and sustainability, such studies are required in order to suggest ways to improve intake among vulnerable populations. It is critical to improve the nutritional status of the elderly, who are more prone to bone fractures. As malnutrition is very prevalent among elders with bone or hip fractures, proper nutrition assessment and intervention are needed.

Another interesting view in maintaining musculoskeletal health is the timing of intake for specific types of nutrients in assuring optimal uptake that benefits its function. For example, the intake of carbohydrates influences insulin release. Insulin influences the transport of tryptophan across the blood brain barrier after each carbohydrate-rich meal, and facilitates the uptake of large neutral amino acids by muscle [129]. According to research, different carbohydrate intake times have different metabolic effects. In accord with carbohydrates, the intake of important nutrients for muscle, such as tryptophan-rich proteins, at different times of the day also showed variability in the body’s response [130]. However, the majority of these studies did not explore the direct impact of the different timings of intake on musculoskeletal functions.

## 6. Conclusions and Future Directions of Research

The musculoskeletal system is the main system that influences daily activities; hence, any disorders or limitations on its function generally affect an individual’s quality of life. We found that growing interest in the individual circadian regulation associated with disease progression has produced various insights into the reciprocal relationships in the field of nutrition [131]. However, most studies on the musculoskeletal system related to nutrition focus on finding the specific type of nutrients or food with potential benefits for health. The concern that is still scarce relates to the type of food consumed at which specific time of day that may have a specific impact on the system. Furthermore, for those who are overweight or obese, losing weight has always been part of the clinical recommendation to address the problem of musculoskeletal disorders. Particularity on the timing of nutritional intake in accordance with circadian rhythm also exerts an effect on calorie intake, which subsequently affects body weight [113]. Given that circadian regulation influences the times at which various physiological processes are optimised, taking into account individual differences in circadian regulation, which recent studies have labelled as “chronotype,” is critical for personalised nutrition recommendations in populations with musculoskeletal disorders. Similar to the approach of chrononutrition, this also applies to nutritional supplements, with the idea that specific supplements may provide better effectiveness during certain times of the day. The specific timing when supplements should be consumed relates to individual sleep and energy patterns [121]. However, the only setback is the lack of conclusive evidence on the specific timing.

The evolution of chronobiological studies has recently left pressing questions to be explored. As described in Figure 1, recommendations on body response are based on the 24 h clock with a specific sunrise time of around 6 am. However, sunrise times vary across the globe; as a result, these differences affect the sleep and wake timing of populations, as well as the onset of dim light melatonin [132]. These variations further affect the best time for daily activities and their relationship with musculoskeletal health. Hypothetically, the body’s 24 h chronological events may be best described in terms of hours and minutes from wake time, rather than being assigned to a specific 24 h time of day. Thus, the body response of those living in regions with different seasons can be better understood. With respect to specific food and supplement intake, this area of study must be further explored to allow effective interventions. Collaborations between nutritionists and physiologists in this chronobiological and chrononutrition field will further clarify these interactions and ultimately lead to successful interventions to reduce morbidity due to musculoskeletal disorders.

## Figures and Tables

**Figure 1 nutrients-15-00734-f001:**
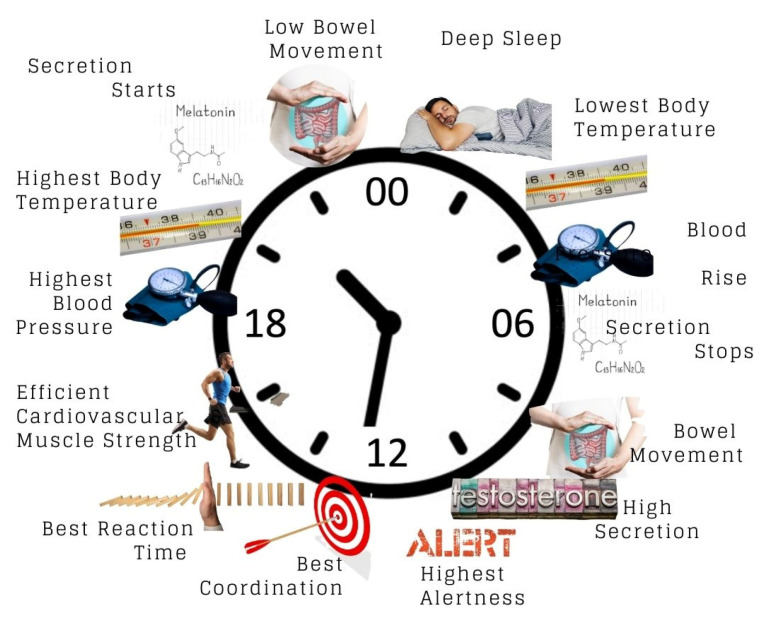
24 h circadian rhythm and human body response. The clock system in Figure 1 represents 24 h clock time that uses the numbers 00:00 (midnight) to 23:59 to tell the time.

**Figure 2 nutrients-15-00734-f002:**
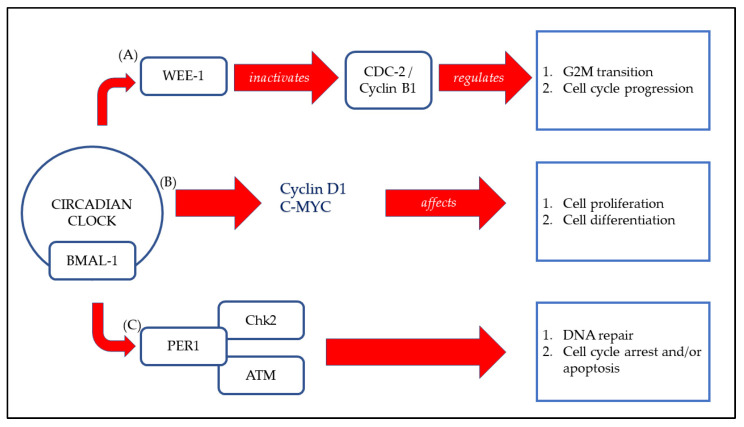
Molecular interaction of cell cycle components with the circadian core clock. (**A**) CLOCK:BMAL1 complex is able to control the transcription of the cell cycle-related gene Wee-1 directly and encodes a protein kinase that inactivates the CDC2/Cyclin B1 complex. Hence, G2-M transition and cell cycle progression are regulated. (**B**) CLOCK:BMAL1 transcriptional activation of the genes encoding Cyclin D1 and *C*-MYC affects cell proliferation and differentiation. (**C**) PER1 complex with the ATM kinase and the checkpoint kinase Chk2 impinges DNA repair, cell cycle arrest, and/or apoptosis [28].

**Figure 3 nutrients-15-00734-f003:**
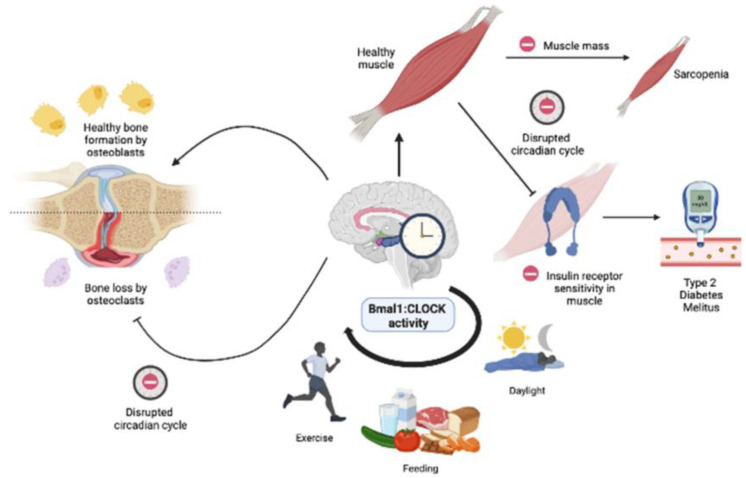
Simplified diagram illustrating the musculoskeletal changes with circadian disruption. The Bmal1: CLOCK activity is regulated by exercise, diet, and sleep quality. Disruptions in the Bmal1: CLOCK can compromise healthy skeletal and muscle function. In healthy muscles, circadian disruption can lead to decreased muscle mass and insulin resistance, which is linked to sarcopenia and type 2 diabetes mellitus, respectively. In bones, circadian disruption shows increased differentiation in osteoclasts leading to bone loss.

**Table 1 nutrients-15-00734-t001:** In vitro, in vivo, and ex vivo works depict that circadian rhythm regulates bone metabolism and bone disease. Circadian rhythm is well known to affect bone metabolism and bone disease. However, its molecular mechanism is still not well understood. The table above shows some of the findings on the molecular mechanism involving circadian rhythm and bone metabolism.

Bone Disorders/Abnormalities	Study Population/In Vivo/In Vitro/Ex Vivo	Circadian Disorders/Disturbance
Bone loss and suppression of bone formation marker	Young adult men [97]	Sleep restriction with concurrent circadian disruption induced a relatively rapid decline in bone formation marker; *N*-terminal propeptide of type 1 procollagen (P1NP) and levels remained lower with ongoing exposure.
Post-menopausal osteopenia	Postmenopausal women [98]	Short-term sleep restriction and circadian disruption can adversely affect bone metabolism, and it was concluded that there is no P1NP recovery with ongoing exposure that, taken together, could lead to lower bone density over time.
Bone turnover marker (BTM) level	Young adult men and women [99]	The diurnal rhythm of bone remodelling is affected by nocturnal dietary patterns.
Femoral neck bone mineral density and content	Shift workers (men and women) age 50 years and above [94]	Intake of a dairy-based protein supplement fortified with calcium at bedtime potentiates nocturnal bone resorption rates in osteopenic postmenopausal women.
Total bone mineral density	Postmenopausal women [100]	*N*-terminal propeptide of type 1 procollagen (P1NP), osteocalcin, and *C*-telopeptide of type 1 collagen (CTX) have no significant effects on sleep restriction, either with or without the opportunity for recovery sleep.
Post-menopausal osteopenia	Postmenopausal women [101]	The effect of sleep restriction on bone metabolism potentially changed with other contributing factors such as age, sex, weight change, and circadian disruption.
Bone misconstruction, early osteoporosis	In vivo animal model; long-term continuous light exposure to mice strain C57BL/6 [83]	Middle-aged male shift workers have a significantly higher total femur and femoral bone mineral content (BMC) than women. Despite having physiological disturbances and hormonal changes due to shift work for years, there are no differences in history of fracture prevalence in men and women.
Bone mineralisation/bone formation	Animal model and ex vivo organ culture; neonatal murine calvarial organ cultures [102]	Sleeping 5 h or less per night was found to have significantly higher odds of low bone mass when compared to 7 h of night sleep. The chances of having osteoporosis of the total hip, spine, and whole body were at higher risk with short sleep duration.
Bone resorption and bone density	Cell and gene knockdown model; osteoblast-specific Bmal1-knockout mice and Bmal1 deficiency osteoblast cell culture [84]	Intake of dietary calcium daily at bedtime results in a significant reduction in biomarkers on homeostatic bone remodelling. The intervention, however, did not change the site-specific bone mineral density or trabecular bone score.
Bone formation in response to mechanical loading	Animal model; sham-loaded mice (in vivo mechanical loading) [103]	Continuous exposure to artificial light disrupts circadian rhythm and affects trabecular density. Data showed that the central circadian rhythm of SCN and the trabecular density decreased, and an inflammatory response was induced. As SCN neurons rebound to normal circadian rhythms, the early osteoporosis gradually recovered.
Implications of circadian oscillators in response to sympathetic nervous system activation	Cell and light–dark cycle animal model; gene knockdown (siRNA transfection) in MC3T3-E1 osteoblastic cells [104]	Bone mineralisation is regulated by the local circadian oscillator signalling pathway, and Per1 expression is involved in the process. However, the regulation of circadian signalling in this mechanism remains vague. Data suggest that circadian signalling plays a role in intramembranous ossification, particularly in the nucleation of apatite minerals. However, the roles of circadian rhythm in mineral propagation and crystallinity are still unclear. Due to this limitation, the current data suggest no influence.
Bone resorption	Cell; osteoclast derived from RAW264.7 [105]	Bone mass and bone resorption are regulated by the osteoblastic circadian clock system. Bmal1-expressed osteoblast, a core component of the circadian clock systems, inhibits the mechanism of bone resorption. Osteoblast-specific deficient Bmal1 results in low bone mass, specifically the lower BMD in the femur and tibia. This shows that Bmal1 inhibits osteoclastogenesis through its expression in osteoblasts. RANKL and osteoclasts increase in Bmal1 knockout mice, while in vitro study shows Bmal1 deficiency does not affect the osteoblast differentiation and maturation. In summary, the deletion of Bmal1 in osteoblasts recapitulates the deletion phenotype. Thus, higher bone formation, higher bone resorption, and lower bone mass. In vitro study shows Bmal1 regulates osteoclastogenesis and bone resorption via RANKL expression in osteoblasts.
Osteoblast regulation	Cell and gene knockdown models; Bmal1 knockout mice with light and dark cycle exposure. [106]	Circadian clocks influence mechanoresponse. This study shows that daily rhythms of clock genes are displayed in bone tissue, and mechanical loading affects circadian rhythm and bone response, and that time of day at which loading is applied affects this mechanism as well.
Osteoclast differentiation and bone mass	Cell and gene knockdown models; Bmal1-specific knockout mice and in vitro osteoclast cell differentiation [59]	Osteocyte mechanoresponsive genes Sost and Dkk1 are differentially regulated based on the time of day, as it moderately affects the response of bone formation in response to the in vivo mechanical loading in mice. This study shows the involvement of circadian genes, and from the zeitgeber data (ZT) of the mechanical loading, ZT14 showed greater endocortical bone formation compared to ZT2. The finding showed that loading time played a role in bone formation.
Osteoblast regulation and bone formation	Cell and gene knockout models; Per2 and Cry2 knockout mice and osteoclast cells [107]	Osteoblast and sympathetic nervous system are involved in bone formation by modulating the core clock genes via the β-adrenergic receptor (β-AR) in osteoblast. This study shows that a transcriptional factor, Nfil3 (a non-selective β-AR agonist) regulates the expression of Ptgs2 involving circadian clock genes Per2 and Bmal2 expression.
Bone formation and osteoblast formation abnormalities	Cell and gene knockout model; Bmal1 knockout mice and L929 cell [88,108]	These data suggest that glucocorticoids are involved in the transmission of the circadian timing from the SCN to peripheral osteoclasts. The osteoclast peripheral clock plays a role in the circadian rhythm and may be regulated by CTSK and NFATc1 expression.
Diabetes mellitus and bone metabolism	Cell and gene knockout model; Goto–Kakizaki rats and bone marrow mesenchymal stem cell [109]	Phosphate metabolism and sympathetic tone are activated during food intake. Phosphate metabolism is regulated by circadian rhythm, and phosphate is an important metabolite for bone metabolism. Data show that fibroblast growth factor 23 (Fgf23) regulates phosphorus levels in osteoblasts. Additionally, skeletal Fgf23 expression levels are high, consistent with urine epinephrine (marker for sympathetic tone) level. In addition, Fgf23 serum levels from mice are low in the daytime but elevated at night. All these data show that circulating Fgf23 level influences the excretion of phosphate in urine, and phosphate metabolism depends on circadian clock network and is relevant to food intake-associated sympathetic activation.
Low phosphate intake and bone healing	Cell and low phosphate-fed animal model [110]	Osteoclast differentiation decreased and bone mass increased in a Bmal1-specific knockout mouse. Cell-based assay and animal study showed that BMAL1 upregulated the calcineurin-dependent 1 (NCATc1) expression by binding to the E-box element of the NFATc1 promoter incorporation to the CLOCK genes. In addition, the steroid receptor coactivator (SRS) family was also involved in the interaction and regulation of BMAL:CLOCK1 transcriptional activity. These findings show that BMAL is highly involved in osteoclast and bone resorption molecular mechanism.
Low back pain, intervertebral disc degeneration	Cell and gene knockdown model; ex vivo intervertebral disc (IVD) explants from (PER2: luciferase (LUC) reporter mice) and human disc cells [111]	Bone formation is significantly increased in Per2 and Cry2 knockout mouse. Bone formation rate and osteoblast biological process are regulated by these genes in distinct pathways, Cry2 predominantly influencing mostly the osteoclastic and Per2 predominantly on the osteoblastic activity.

## Data Availability

All data are available within the article and publicly accessible.

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
