# Peer review of "Effect of Circadian Rhythm Disturbance on the Human Musculoskeletal System and the Importance of Nutritional Strategies"

_nutrients, 2023, doi:10.3390/nu15030734_

Round 1

Reviewer 1 Report

The review manuscript ‘Effect of Circadian Rhythm Disturbance on the Human Musculoskeletal System and the Importance of Nutritional Strategies’ is a well-written manuscript with relevant information and references to demonstrate the previous studies on how circadian rhythm disruption contribute to the effects on the musculoskeletal system. In the introduction part, authors firstly described the physiological function of central and peripheral circadian clock and how clock genes control the circadian system working synchronously. In addition, authors described that the circadian rhythms in skeletal muscle and bone functions which are also the main focuses of the manuscript in the review part. In the main part of the manuscript, authors firstly reviewed the previous studies on human circadian rhythm and how the circadian clock are biologically regulated and measured; how the circadian rhythm affects different body systems especially maintaining hormones and metabolism. Next, authors reviewed the previous studies using genetic model and revealed that circadian dysfunction has a great impact on Musculoskeletal System, lipid and glucose metabolism. In the next part, authors also reviewed the previous studies on how circadian rhythm regulating bone metabolism and bone health. In the last part, authors reviewed and suggested that how nutrition such as vitamin D and calcium maintain and benefit bone and muscle health especially to the elderly population. The manuscript is outstanding and informative. 

Reviewer 2 Report

The article is very interesting, it regards the disturbance of the circadian rhythm and its resultant effects on the musculoskeletal system.

There are some misleading aspects or insufficient data on some subjects that need further attention and correction. Figure 1 is an uncommon representation of time that can lead to confusions.

The descriptive biochemical mechanism encoding circadian rhythms should be depicted in a figure or schematic representation. 

Regarding calcium and vitamin D theme, a discussion should be made on the nutritional strategies in conjunction with circadian (mis)alignment. You should emphasize the importance of proper natural light exposure for children and nevertheless for adults, in order to assure and sustain a physiological concentration of vitamin D, and for establishing the balance and activate the modulatory role of light-dark alternation in 24 hours time.

In my opinion you should expand the chrononutritional details presented from 440 to 449, in the respective paragraph, in a separate subchapter. 
